# The Aerodynamic Effect of Biomimetic Pigeon Feathered Wing on a 1-DoF Flapping Mechanism

**DOI:** 10.3390/biomimetics9010036

**Published:** 2024-01-05

**Authors:** Szu-I Yeh, Chen-Yu Hsu

**Affiliations:** Department of Aeronautics and Astronautics Engineering, National Cheng Kung University, Tainan City 701, Taiwan

**Keywords:** rock pigeon, feathered wing, flapping mechanism, artificial remiges, stiffness

## Abstract

This study focused on designing a single-degree-of-freedom (1-DoF) mechanism emulating the wings of rock pigeons. Three wing models were created: one with REAL feathers from a pigeon, and the other two models with 3D-printed artificial remiges made using different strengths of material, PLA and PETG. Aerodynamic performance was assessed in a wind tunnel under both stationary (0 m/s) and cruising speed (16 m/s) with flapping frequencies from 3.0 to 6.0 Hz. The stiffness of remiges was examined through three-point bending tests. The artificial feathers made of PLA have greater rigidity than REAL feathers, while PETG, on the other hand, exhibits the weakest strength. At cruising speed, although the artificial feathers exhibit more noticeable feather splitting and more pronounced fluctuations in lift during the flapping process compared to REAL feathers due to the differences in weight and stiffness distribution, the PETG feathered wing showed the highest lift enhancement (28% of pigeon body weight), while the PLA feathered wing had high thrust but doubled drag, making them inefficient in cruising. The PETG feathered wing provided better propulsion efficiency than the REAL feathered wing. Despite their weight, artificial feathered wings outperformed REAL feathers in 1-DoF flapping motion. This study shows the potential for artificial feathers in improving the flight performance of Flapping Wing Micro Air Vehicles (FWMAVs).

## 1. Introduction

In the realm of natural flight organisms with flapping wings, they can primarily be categorized into insects with membranous wings, birds with feathered wings, and bats with membranous wings that intermediate between the two [1]. Among these, birds’ feathered wings consist of multiple layers of overlapping feathers [2]. They possess the unique capability of articulating and adjusting wing shape in conjunction with joint movements. Furthermore, certain flight feathers exhibit unidirectional airflow, enhancing the efficiency of lift and thrust generation during flapping. This adaptability enables birds to employ the most efficient modes of motion and flight strategies to accommodate various flight conditions and purposes [3,4]. Previous studies have predominantly relied on methods such as biological observation [5,6], numerical simulations [7,8], and scale-model wind tunnel experiments [9,10,11]. While birds may not match the agility of smaller insects, their larger size grants them access to higher flight altitudes, greater speeds, increased payload capacity, and extended ranges. Consequently, birds’ flight has remained a pivotal focal point in the field of biomimetic flapping flight research.

From previous research, it is evident that the aerodynamic force generation in flapping wings is influenced by factors such as geometric shape, motion patterns, and deformation modes. This influence is particularly pronounced in thrust generation, where the elasticity and deformation of the wing surface play a pivotal role [12]. Both numerical simulations and experimental measurements have consistently demonstrated that wings with high rigidity generate significantly lower thrust during flapping motion compared to wings with flexibility [13,14,15,16]. However, excessively low wing rigidity can also lead to suboptimal thrust production. Thus, achieving an optimal distribution of wing rigidity is a significant challenge in the design of flapping wings. Among the various flight organisms in the natural world, birds’ wings exhibit the most complex functionality. The primary flight feathers located on the inner and outer wing surfaces collectively contribute to the majority of the wing area. They exert a decisive influence on the distribution of wing rigidity, fluid-structure coupling phenomena, and aerodynamic force generation.

To gain a deeper understanding of the impact of different feather rigidities on flight performance, wind tunnel experiments with model wings featuring fixed motion patterns and geometric shapes, but varying feather rigidity, can be conducted to mimic real birds’ flight conditions. This approach holds the potential to provide valuable insights into the effects of different feather rigidities on flight performance. The researchers had emulated the flapping pattern of a rock dove (pigeon) flying at 10 m/s wind speed. They designed a single-degree-of-freedom (1-DoF) biomimetic flapping mechanism composed of a carbon fiber skeleton and plastic membranes for both the inner and outer wing surfaces. Their experimental results revealed that the wing surfaces with passive openings during the upstroke, as opposed to fully enclosed wing surfaces, exhibited superior lift and thrust generation [17]. In recent years, drones composed of flexible wing surfaces not made of inflexible elements have been gradually developed. Among these, the Morphing-Wing-Based Gliding biomimetic aircraft with feathered wings is one such direction of development [18].

Current research on artificial feathers predominantly focuses on properties such as waterproofing, resistance resulting from surface structures [19,20], and optical performance [21]. There is limited research on the aerodynamic performance of artificial feathers, and most studies involve their direct application on micro aerial vehicles. Moreover, several research teams have begun incorporating the concept of artificial feathers into the design of flapping-wing aircraft. In 2021, Festo Corporation introduced the BionicSwift, a biomimetic flapping-wing aircraft constructed using carbon fiber composite materials for the main wing framework and shafts, with lightweight foam material feathers adhered to the shafts [22]. This combination creates wings that closely resemble those of real birds. The carefully designed wing components exhibit sufficient rigidity, allowing them to achieve remarkably lifelike wing deformation and feather movement with just a single degree of freedom in the flapping motion. The PigeonBot, a biomimetic winged aircraft developed by a research team at Stanford University, utilizes pigeon feathers overlaid onto a 3D-printed biomimetic jointed skeletal structure [23,24]. This construction results in a fixed-wing biomimetic aircraft capable of altering its wing planform geometry. In addition to mimicking the rock dove’s ability to adjust wing shape for varying lift-to-drag ratios, it also employs asymmetric wing folding motions for roll control, successfully replicating certain aspects of avian wing functionality in flight.

The main objective of this study is to further explore the influence of different feather rigidities on the aerodynamic characteristics of flapping wings. To achieve this, a 1-DoF flapping mechanism was designed to mimic the size and shape of a rock pigeon. The rigid main structure of the wing model was designed to correspond to the skeletal, muscular, and skin tissues found in bird wings. Flexible feathers were overlaid on the trailing edge of the main structure to create a realistic flexible wing surface. In the wind tunnel, tests were conducted under conditions simulating the flapping-glide flight mode of a rock pigeon, including wind speeds and motion patterns. Throughout the entire flapping cycle, measurements were recorded for changes in lift, thrust, and the deformation state of the feathers. To investigate the aerodynamic effects of varying feather rigidities, three sets of feathers were prepared: one with high rigidity, one with low rigidity, and one set obtained from real rock pigeons. These feathers were attached to the main structure of the wing model for wind tunnel experiments. By comparing the aerodynamic performance of these three sets of feathers with different rigidities, we aim to provide valuable insights into the possibilities for the design of Flapping Wing Micro Air Vehicles (FWMAVs) through this research on the 3D-printed artificial feathers.

## 2. Experimental Setup

### 2.1. Flapping Mechanism

According to the literature [4], when rock pigeons engage in steady-level flight, their flapping frequency is approximately 6 Hz under the experimental airspeed conditions (6–20 m/s). The angle between the stroke plane of the wings and the horizontal plane is approximately 90° at airspeeds between 16 m/s and 20 m/s. At lower speeds, this angle decreases, with an average flapping angle of 93°. Furthermore, the wrist joint movement becomes more concentrated at flight speeds exceeding 12 m/s, with only the wingtips folding backward during the upstroke. When flying at 16 m/s, the duration of the upstroke is the same as that of the downstroke. At higher speeds, the proportion of time spent on the downstroke decreases, while at lower speeds, there is a tendency to increase the duration of the downstroke. In summary, when flying at a steady speed of 16 m/s, rock pigeons exhibit a relatively simple flight pattern. Apart from the folding of the outer wing during the upstroke, their wing motion is predominantly a single-degree-of-freedom (1-DoF) flapping motion with the shoulder joint as the axis. The purpose of this study is to investigate the aerodynamic benefits of different feather rigidities. Consequently, other factors contributing to aerodynamic effects are treated as control variables, and experimental design is simplified accordingly to isolate the effects of feather rigidity on the aerodynamic performance of flapping wings. To meet these experimental requirements, this study selects the condition of continuous flapping flight at 16 m/s, observed in rock pigeons; the flapping mechanism adopts a design characterized by a 1-DoF shoulder joint, symmetric flapping angles, symmetric motion patterns, and symmetric wing geometry.

The mechanism designed for this study, along with its corresponding coordinate system, is depicted in Figure 1a. The x-axis is defined as facing the direction of freestream, the y-axis as the horizontal wing span direction of the left wing, and the z-axis as the direction of flight lift generation. To convert the constant rotational motion of the driving motors into the desired flapping motion according to the experimental design, the flapping mechanism in this study employs cylindrical cam-driven transmission pins to perform linear reciprocating motion. This reciprocating motion simultaneously drives the rotation of the two-wing slider rocker assemblies, completing the flapping motion. The overall mechanism is illustrated in Figure 1b, with each of the two wing surfaces being driven using a cylindrical cam. Beneath the cam, transmission gears are installed, driven synchronously using the same dual-stage reduction gear, ensuring symmetrical movement of both wing surfaces. Behind the dual-stage reduction gear is a cycloidal drive gearbox. For detailed mechanical design and operational mechanisms, please refer to the Appendix A. The components of the mechanism are primarily fabricated using PLA material through 3D printing. Due to the higher loads and operating temperatures, the inner and outer gears of the cycloidal drive gearbox are manufactured using aluminum alloy wire cutting methods.

The flapping motion is defined as shown in the front view in Figure 1c, with the left shoulder joint as the axis (parallel to the x-axis). The angle between the xy-plane and the wing’s central plane is defined as the flapping angle (*ϕ*), with upward rotation considered positive. The mechanism’s operational mode is designed to operate in an up-down symmetric manner, with the shoulder joint axis parallel to the x-axis. The distance between the two shoulder joint axes (*w*) is 60 mm. The angular displacement is defined as shown in Equation (1), with a flapping motion amplitude (Φ) of 45° and the frequency (*f*). The maximum achievable flapping frequency within the operational constraints of the mechanism is 6 Hz which fits the flapping frequency observed in rock pigeons flying at 16 m/s airspeed.
(1)∅=Φsin⁡(2πft)

### 2.2. Feathered Wing Models Design

In this study, the reference of the feathered wing model was made to 3D scans of rock pigeon wing specimens using previous research [25]. This data was used to establish the three-dimensional geometry and parameters of the wings, including geometric parameters for each wing segment such as chord length, thickness-to-chord ratio, and maximum camber position. Based on this experimental data and the comparison of sample sizes collected for this study, the geometric layout of the model wing was constructed, as illustrated in Figure 2a. The half-wing length (*R*) is 327.5 mm, the half-wing area is 320 cm^2^, and the half-wing aspect ratio is 3.35. Considering the width of the body, the combined wingspan (*b*) is 737 mm, the total wing area is 742.5 cm^2^, and the total aspect ratio is 7.32. The wing’s planform is based on the aforementioned parameters, the leading-edge boundary was delineated, and the angles and positions for mounting the flight feathers were determined. Furthermore, based on the thickness and chord length distribution data of the rock pigeon wing specimens, symmetric NACA four-digit wing sections with a thickness distribution similar to that of the specimens were drawn starting from the wing root and progressing at 5% increments along the wing length. The wing’s trailing edge boundary was defined based on the point where the feathers are embedded, resulting in a wing outline with a similar thickness-to-chord ratio to a real pigeon wing but lacking camber, as shown in Figure 2b. This geometric design of the remiges is employed in this study.

The wing structure designed for this study, capable of replacing the artificial remiges/feathers, is show in Figure 2c. The wing shell is fabricated using PLA material through 3D printing and is divided into upper and lower wing panels. Both upper and lower wing panels feature a total of 17 feather slots labeled as P1–P10 and S1–S7, named based on the biological numbering system for feathers. These feather slots are designed to accommodate the shapes of both artificial and real feathers. The bottom surface of the feather slots gradually rises from the distal end to the proximal end in 0.15 mm increments. This design ensures that when feathers are installed, they stack in the order of distal feathers at the bottom and proximal feathers at the top, forming a smooth and continuous wing surface. This structure allows for the assembly of three types of flapping wings with different feather mechanical properties: low rigidity and high rigidity artificial biomimetic feathers, as well as real feathers obtained from biological samples.

The REAL (Remiges Extracted from Avian Limbs) feather samples, obtained from the same retired racing pigeon carcass, consist of a total of forty feathers from both the left and right wings, labeled with markings at the juncture where each feather enters the skin. Measurements were taken for the width and height at this juncture, and top-view photographs were captured as modeling references. Both types of biomimetic feathers were produced using 3D printing technology. For biomimetic feather modeling, the right-wing feather of the previously mentioned model was used to outline its planar geometry. Geometric modeling of the feather shaft section involved establishing reference planes at the proximal end, distal end, and the juncture between the shaft and the feather vane. The cross-sectional geometry featured differently sized upright ellipses.

Modeling of the feather vane’s planar geometry utilized photographs of the biological sample to outline the top-view geometry of the feather vane section, which is in the same plane as the bottom surface of the feather rachis. The cross-sectional geometry of the rachis was achieved through establishing reference planes at the proximal end of the rachis, the distal end of the rachis, and the junction of the rachis and the vane. The cross-sectional shape was set as an upright half ellipse, and the distal end of the feather shaft was a semicircular shape with a diameter of 0.3 mm. The three reference planes were constructed with tangency rather than curvature, using the midline of rachis as the reference axis to create a multi-section solid. The bottom surface of the vane was aligned with the rachis as a coplanar plane, then elevated in a pad with a thickness of 0.15 mm for 3D-printing. The artificial feathers were constructed using two materials, PLA and PETG. This choice was made due to the similar densities and tensile strengths of these two materials, which closely match Young’s modulus of real feathers [26,27,28]. This allowed for the creation of experimental samples with the same shape and weight as real feathers but with varying degrees of rigidity, both higher and lower than real feathers. The assembled wing, with REAL feathers on the right-wing surface as shown in Figure 2a and PLA and PETG biomimetic feathers sequentially inserted, but not yet covered by the upper and lower wing panels, is depicted in Figure 2d,e.

Considering the descriptions of these two sections and referencing the dimensions of the wing and the motion kinematics of real pigeons, the relevant parameters for the mechanism and wing model used in this study are listed in Table 1. After the assembly of the mechanism and feathers was completed, multiple cameras were used to observe the variation in flapping angles of the mechanism. The correlation coefficient for each model with the designed motion are all above 0.98, confirming that the mechanism accurately performs the intended flapping motion in all experiments.

### 2.3. Experimental Setup

The wind tunnel experiments of this study were conducted in the closed-circuit low-speed wind tunnel at the Aerospace Science and Technology Research Center of National Cheng-Kung University. The test section dimensions were 2750 mm (length) × 1850 mm (width) × 1200 mm (height), with a wind speed range of 0 m/s to 65 m/s. The turbulence intensity within this range was less than 0.5%, ensuring good and stable wind field quality [29]. In the wind tunnel experiments, five 5 kg-class single-point load cells were used to measure the vertical forces (lift) and the horizontal forces (thrust). The experimental setup is shown in Figure 3. The horizontal force acting on the mechanism was measured using a load cell directly anchored to the bottom plate of the flapping mechanism frame and the wind tunnel column. The vertical force was measured using a platform composed of four load cells, anchored between the wind tunnel baseplate and the tunnel wall mounting bracket. Each set of wing models underwent the same wind tunnel test parameters and identical flapping motions. Measurements were conducted at both 0 m/s (static state) and 16 m/s (flight speed). The flapping mechanism was accelerated to its maximum flapping frequency of 6 Hz, and the experiments lasted for a total of 12 s. Data averaging was performed over different flapping cycles after excluding the first and last flapping cycles to investigate the variation in forces with flapping motion.

## 3. Results and Discussion

### 3.1. Stiffeness Distribution of the Feathers

In this study, a non-destructive testing method was employed to measure the bending stiffness of both REAL feathers and biomimetic feathers. A three-point bending test [30] was conducted at various points along the feather shaft. The feather shaft’s proximal end, near the attachment point to the model, served as the starting point, while the distal end, which is the tip of the feather shaft, served as the endpoint. Five equally spaced test points were selected in between, and the bending stiffness (*EI*) at different positions along the feather shaft was measured using the three-point bending test. The stiffness at the tip of the feather shaft was defined as zero, and linear extrapolation was performed to obtain the distribution of bending stiffness along the feather shaft’s length. The average bending stiffness ($\bar{EI})$ of the feather sample was then calculated based on the definition in Equation (2). P3, P6, and P9 remiges were selected for bending tests, and measurements were conducted on both the left and right wings of REAL feathers. Additionally, a bending stiffness parameter (*λ*) was defined based on the dynamic equation of a cantilever beam-like aeroelastic system, considering the flapping conditions of the mechanism, to investigate the influence of material properties on feather structure, as described in Equation (3).
(2)$$(\bar{EI})=\frac{\int EI\left(s\right)ds}{Lf}$$
(3)$$\lambda =\frac{\bar{EI}}{\rho \Phi^{2}R^{2}f^{2}b^{4}}$$

The results of the stiffness tests for the three types of feather samples, namely PLA, PETG, and REAL feathers, are presented in Figure 4. For the artificial feathers, the highest bending stiffness for PLA feathers is located at approximately one third of the feather shaft length, and the stiffness distribution is significantly higher than or roughly equal to the other two materials. PETG feathers exhibit lower overall stiffness distribution, but stiffness starts to become slightly higher than that of real feathers at positions beyond half of the feather shaft length. The bending stiffness distribution for REAL feathers on both left and right wings is nearly identical. The results show that the high stiffness region is concentrated in the front half of the feather shaft, closer to the base region, where the bending stiffness of the feather shaft is significantly higher than the other two types of artificial biomimetic feathers. However, the structure at the tip of the feather shaft is extremely soft. Overall, PLA feathers have the highest average bending stiffness, followed by REAL feathers, while PETG feathers are the softest, still fitting the design purpose of this study. Furthermore, Figure 4 also presents the calculated bending stiffness parameters (*λ*) for the three feather sample types, with the reference airspeed based on the average wingtip velocity, considering the air density of 1.146 kg/m^3^ on the experimental day, and these results are also consistent with the feather design objective.

Figure 5 shows the images captured at the maximum deformation during both the upstroke and downstroke movements under a free-stream velocity of 0 m/s and a flapping frequency of 6 Hz. These images represent the moments when the upstroke and downstroke movements conclude and are about to transition. Comparing the deformation of the three feathered wing models, it is observed that the deformation magnitude decreases in the order of PETG, PLA, and REAL feathered wings. Additionally, during both the upstroke and downstroke phases, vibrations in both the out-of-plane and in-plane directions of the wing surfaces are observed. During the downstroke, the aerodynamic forces tend to bring the feathers closer together, resulting in a closed wing surface. Conversely, during the upstroke, the feathers exhibit slight tilting due to the greater bending amplitudes of the distal feathers compared to the proximal ones. Because the bending stiffness distribution of PLA and PETG feathers is relatively similar along the feather shaft, their deformations during flapping do not concentrate on specific region, and the feathers show gradual shape changes. On the other hand, REAL feathers exhibit distinct behavior. Since the bending stiffness is highest at the proximal end and the feather shaft’s tip is extremely flexible, the deformation and vibrations during flapping predominantly occur at the distal end of the feathers. This is a notable difference between the artificial feathers and the REAL feathers.

### 3.2. The Variations in Lift and Thrust

The actuation method used in this study for the flapping mechanism involves oscillatory angular velocity motion with respect to the x-axis. The vertical inertial force during flapping can be calculated by differentiating the shoulder joint angular displacement data measured during the flapping tests. This is achieved by considering the total mass, center of mass position, and eccentricity of the wing model and the mechanism arm. Finally, according to the principles of kinematics, the vertical component of the inertial force can be calculated. The force variation during a flapping cycle was calculated using the average over ten flapping cycles. The solid line represents the mean value, and an error band is drawn in a lighter shade of the same color, with an error band width of ±1 S.E.M.

Figure 6 shows the lift variation calculated via subtracting the vertical inertial force from the measured vertical force. At a flow velocity of 0 m/s, the lift and vertical force profiles for all three feathered wings are similar. The areas of positive and negative lift regions are approximately equal, resulting in an average lift close to zero. During the wing stroke reversal phase, all three models experience a peak in lift force in the opposite direction of the inertial force, indicating that the significant inertial forces during stroke reversal are counteracted by the generated lift. During the phase of maximum wing stroke angular velocity (midway through the upstroke and downstroke), positive and negative lift peaks are generated. At a flow velocity of 16 m/s, the PLA feathered wing exhibits a slight increase in the peak of lift, while the PETG feathered wing experiences a significant flattening of the peak of lift. In the case of the REAL feathered wing, the peak in positive lift (end of the downstroke) disappears, and the peak in negative lift (end of the upstroke) remains roughly the same. In all three wing models, the positive lift region becomes wider and the peak increases compared to static flapping, while the negative lift region becomes narrower, and the peak decreases.

Figure 7 shows the variation of thrust over a flapping cycle. The thrust values for all three wing models increase with the flapping frequency. The PETG feathered wing exhibits the most pronounced thrust oscillations, with a significantly higher peak of lift value but also a larger negative thrust peak and wider range of negative thrust. The PLA feathered wing exhibits thrust oscillations similar to the PETG wing but with smaller amplitude. During the transition phase between the upstroke and downstroke of the REAL feathered wing, the thrust values are lower, and they mostly remain in the positive thrust state throughout the flapping cycle. Even when negative thrust occurs, its magnitude and duration are smaller compared to the other two models. As a result, the REAL wing, while having lower overall thrust output compared to the other two, shows a relatively stable thrust profile.

The average of the vertical aerodynamic forces in a whole flapping cycle, as shown in Figure 8, reveals that under stationary flow conditions (0 m/s), all three feathered wings generate limited average lift. The PLA feathered wing even produces approximately −20 gw of negative lift, indicating that none of the wings generate effective net lift. In the 16 m/s airflow, the REAL feathered wing, owing to the inherent curved shape of pigeon feathers, generates a substantial initial lift of 523.5 gw in an unflapped, steady flow condition. In contrast, both sets of biomimetic wings with artificial remiges generate only around 200 gw of initial lift. With an increase in flapping frequency, all three models experience an enhancement of lift. However, as the flapping frequency further increases, the net lift of all three wing models begins to decrease. Among them, the REAL feathered wing exhibits a more pronounced decline, especially at the 6 Hz flapping frequency which is similar to that of real pigeons. In fact, the PETG feathered wing consistently shows significant increases in lift within the tested range, making it the most effective model in generating net lift among the three feathered wings.

Average thrust is calculated via subtracting the initial drag at rest from the average drag during flapping, and the results are shown in Figure 9. In the absence of incoming flow, the static thrust performance from largest to smallest is in the order of PETG, PLA, and REAL. However, in dynamic thrust at a flow velocity of 16 m/s, the PLA feathered wing shows the highest thrust, while the REAL feathered wing shows the lowest thrust. Notably, for both PLA and REAL wings, the dynamic thrust values aret almost identical to their static thrust at 6 Hz flapping frequency, and for flapping frequencies below 6 Hz, their dynamic thrust is higher than static thrust. Considering the different initial drag values among the three wing models, even though the PLA feathered wing shows the best dynamic thrust performance, the horizontal net force is still less than zero until the flapping frequency increases to 6 Hz. In contrast, PETG and REAL wings generate positive horizontal net forces, resulting in effective propulsion.

### 3.3. The Power Consumption of the Flapping Motion

Figure 10 shows the recorded electrical power consumption during the flapping process of the mechanism. For all three wing models, the power consumption increases at a flow velocity of 16 m/s compared to 0 m/s. Among them, the REAL feathered wing shows the most significant increase, followed by PLA, while PETG shows only a slight increase. Due to the lower total mass and eccentricity of the REAL feathered wing compared to the biomimetic wings with artificial regimes, the REAL feathered wing has the lowest power consumption in both flow conditions, while the PLA and PETG wings have nearly identical power consumption at 0 m/s.

In the experimental design of this study, the REAL feathered wing exhibits the lowest thrust-to-power ratio (shown in Figure 11). However, within the experimental flapping frequency range, the thrust-to-power ratio for the REAL feathered wing increases with the flapping frequency and approaches the thrust-to-power ratios of the other two wing models at 6 Hz. For PLA and PETG wing models at 0 m/s flow conditions, their thrust-to-power ratios also increase with increasing flapping frequency, but the PLA feathered wing shows a smaller increase and a subsequent decrease after 5 Hz, while the PETG wing exhibits a decrease after 5.5 Hz. In the 16 m/s flow conditions, both PLA and PETG wings experience a rapid decrease in thrust-to-power ratio with increasing flapping frequency.

In terms of static thrust, the PETG feathered wing undoubtedly exhibits the best thrust-to-power ratio and static thrust. However, under 16 m/s flow conditions, although the PLA feathered wing has the highest thrust-to-power ratio and dynamic thrust, its initial resistance is about twice that of the other two wing models, and the higher dynamic thrust efficiency is not sufficient to offset this disadvantage. While the REAL feathered wing has the smallest initial resistance and the lowest power consumption among the three wing models, its relatively low thrust-to-power ratio means that it does not perform as well as PETG wings in terms of overall thrust efficiency, considering all test conditions conducted in this study. Thus, the PETG wing emerges as the wing model with the best overall propulsion efficiency.

## 4. Conclusions

In this study, a fabrication process of 3D-printed artificial feathers has been successfully developed. Subsequently, these artificial feathers were subjected to both three-point bending tests and wind tunnel experiments to analyze and compare the structural and aerodynamic performance differences between the wing models mounted with artificial feathers and REAL feathers. Aerodynamic forces generated via artificial remiges exhibit notable fluctuations, whereas REAL wings consistently produce relatively small yet smooth output forces throughout the entire flapping cycle. However, the feathered wing with PETG artificial feathers/remiges can achieve a lift enhancement of up to 28% of body weight. It also exhibits the highest static thrust and can overcome the initial drag in the freestream flow field. In summary, PETG artificial feathers/remiges outperform the other wing models in terms of aero-force enhancement. Although the artificial remiges still fall short of REAL remiges in terms of weight, they show promise in enhancing aerodynamic forces during 1DoF flapping motion. Although the reasons for the varying lift and thrust performances of artificial feathers with different rigidities remain to be further analyzed and studied in the investigations of the induced flow field, this still makes it highly feasible to manufacture biomimetic wings for artificial flying vehicles in the future using simplified manufacturing methods, such as 3D printing. This advancement lays the foundation for developing more sophisticated FWMAVs. The results of this research provide valuable insights and greater design possibilities for future FWMAVs.

## Figures and Tables

**Figure 1 biomimetics-09-00036-f001:**
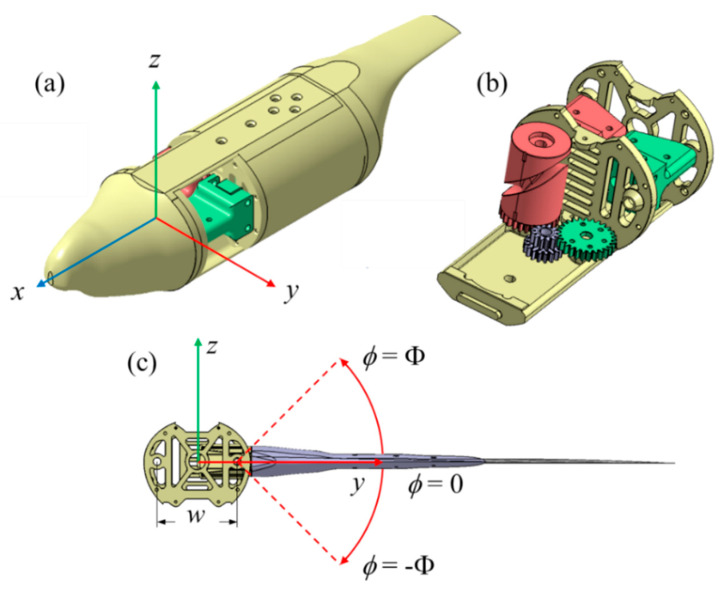
(**a**) Mechanical system overview and coordinate system. (**b**) Internal drive mechanism design within the fuselage. (**c**) Front view of the mechanism and definition of the flapping angle.

**Figure 2 biomimetics-09-00036-f002:**
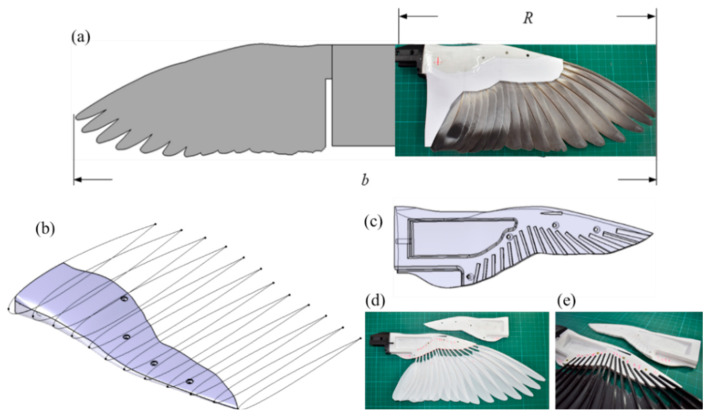
(**a**) The wing shape and the definition of geometric parameters. (**b**) Three-dimensional wing geometry and cross-sectional geometry of each segment. (**c**) The wing shell design at the wing-to-feather connection point. (**d**,**e**) Wing surfaces with attached PLA and PETG artificial feathers, respectively.

**Figure 3 biomimetics-09-00036-f003:**
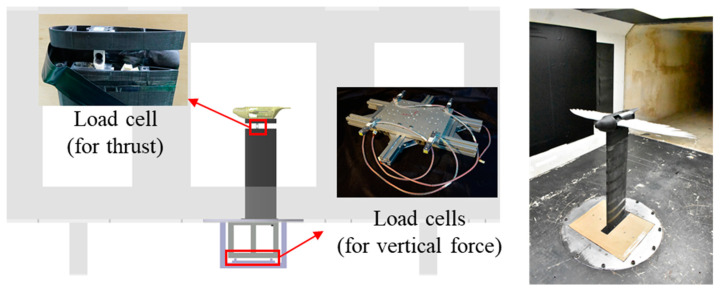
Schematic diagram and pictures of the experimental setup and force measurement device. The entire flapping mechanism is installed and tested in a closed-circuit low-speed wind tunnel. The relative positioning of the model installation and the wind tunnel is also depicted in the schematic diagram. The background in the schematic represents the test section of the wind tunnel and the relative positioning of the observation window.

**Figure 4 biomimetics-09-00036-f004:**
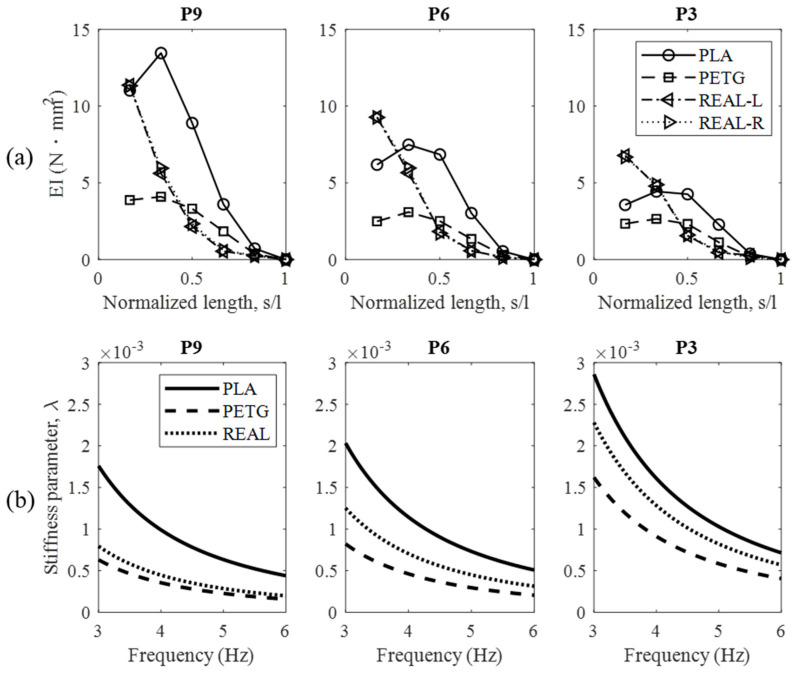
(**a**) The stiffness distribution of P9, P6, and P3 remiges made from different materials and REAL feathers. (**b**) The bending stiffness parameters of P9, P6, and P3 remiges under various flapping frequencies.

**Figure 5 biomimetics-09-00036-f005:**
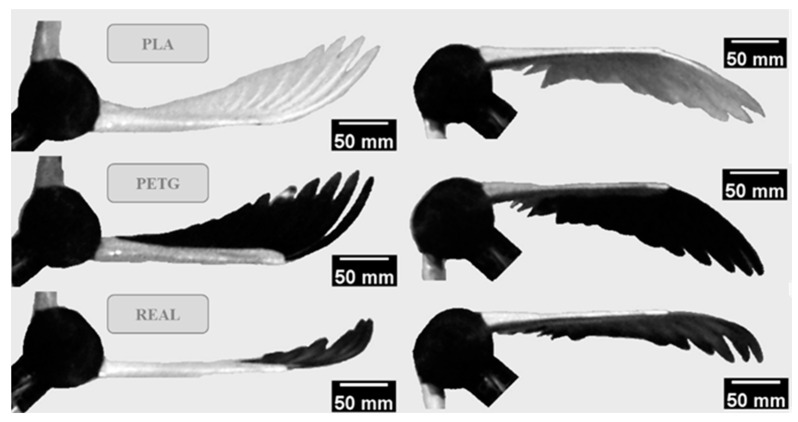
The pictures of maximum deformation of different feathered wings during the upstroke (**left**) and downstroke (**right**).

**Figure 6 biomimetics-09-00036-f006:**
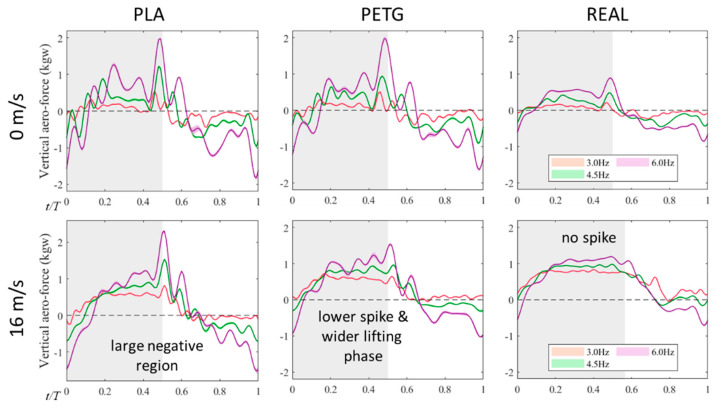
The variation of vertical aerodynamic forces (lift) over one flapping cycle, including the experimental measurements of three different wings with different materials in a stationary state and at the real flight speed (16 m/s).

**Figure 7 biomimetics-09-00036-f007:**
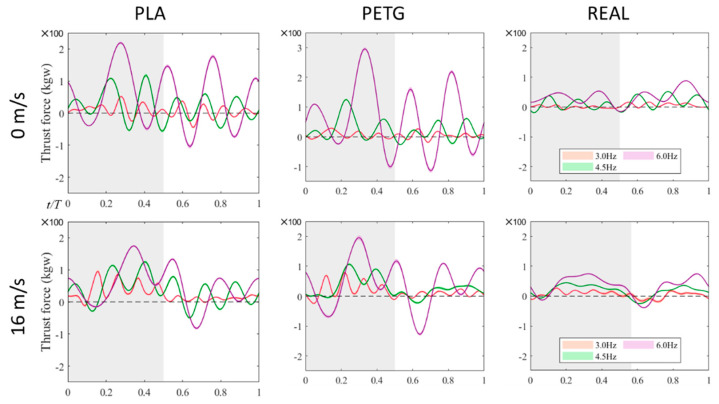
The variation of thrust over one flapping cycle, including the experimental measurements of three different wings with different materials in a stationary state and at the real flight speed (16 m/s).

**Figure 8 biomimetics-09-00036-f008:**
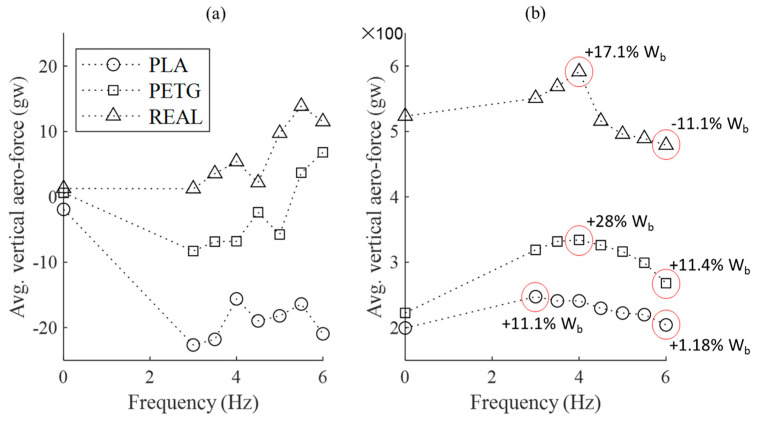
The average lift values over one complete flapping cycle for the three wing models at different wind speeds and flapping frequencies. (**a**) No incoming flow velocity (static); (**b**) Incoming flow velocity of 16 m/s.

**Figure 9 biomimetics-09-00036-f009:**
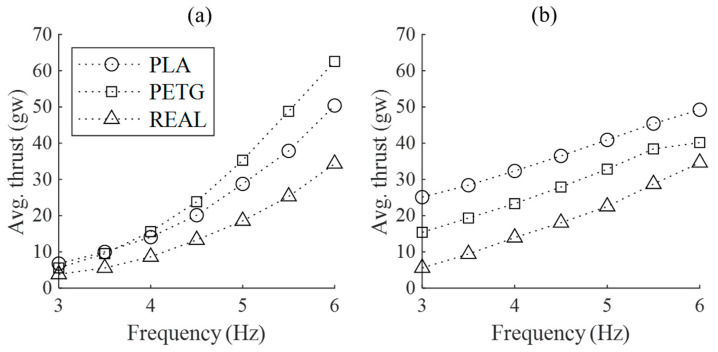
The average thrust values over one complete flapping cycle for the three wing models at different wind speeds and flapping frequencies. (**a**) No incoming flow velocity (static); (**b**) Incoming flow velocity of 16 m/s.

**Figure 10 biomimetics-09-00036-f010:**
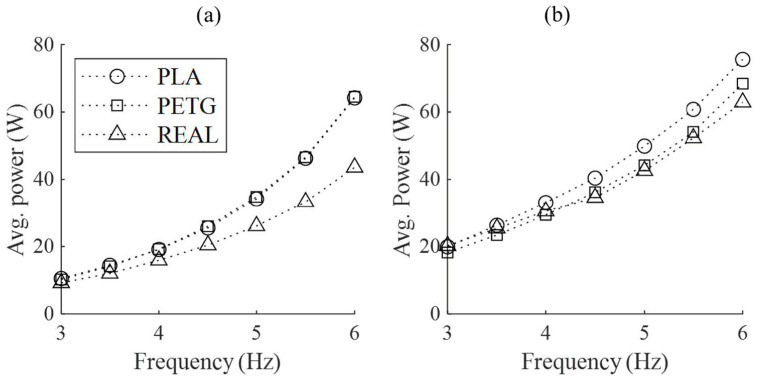
The average power consumption over one complete flapping cycle for the three wing models at different wind speeds and flapping frequencies. (**a**) No incoming flow velocity (static); (**b**) Incoming flow velocity of 16 m/s.

**Figure 11 biomimetics-09-00036-f011:**
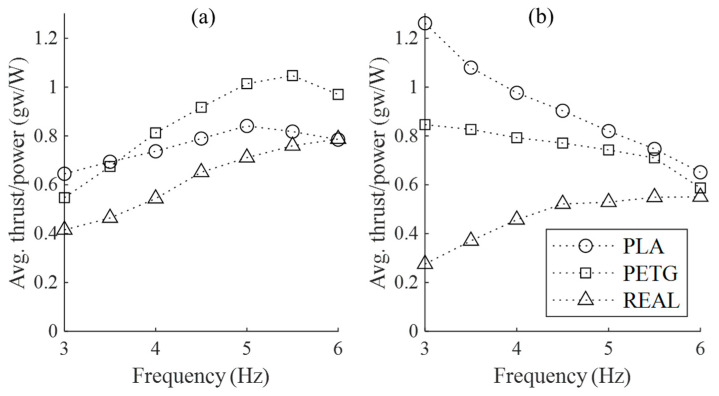
The average thrust-to-power consumption ratio over one complete flapping cycle for the three wing models at different wind speeds and flapping frequencies. (**a**) No incoming flow velocity (static); (**b**) Incoming flow velocity of 16 m/s.

**Table 1 biomimetics-09-00036-t001:** Design Parameters for the Mechanism and Wing Model.

	Real Pigeon [4]	This Study
Wind speed (m/s)	16	0 & 16
Flapping frequency (Hz)	5.7–6	6
Flapping amplitude	~46.5°	45°
Stroke plane	~90°	90°
Down stroke percentage	50%	50%
Wing span (cm)	62–72	73.7
Avg. wing length (cm)	28–32.5	32.75

## Data Availability

The data that support the findings of this study are available from the corresponding author upon reasonable request.

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
