# Peer review of "The Aerodynamic Effect of Biomimetic Pigeon Feathered Wing on a 1-DoF Flapping Mechanism"

_biomimetics, 2024, doi:10.3390/biomimetics9010036_

Round 1

Reviewer 1 Report

Comments and Suggestions for Authors

General Comments:

This paper aimed to analyze the effect of different feather rigidities on the aerodynamic characteristics of flapping wings. This study may provide indispensable information for improving the flight ability of FWMAVs. However, there are shortcomings in the research design, incomplete research content, and issues with experimental design, data analysis, and writing details. I am not sure the paper can be suitable for publication even after a major revision.

Detailed Comments:

1.       The effect of wind speed was not reflected as the measurement was only conducted at two velocities of 0 m/s and 16 m/s. Will high wind speeds cause damage to different types of wings?

2.       The force testing is not comprehensive as it did not measure the lateral force and torque. As a result, it is difficult to evaluate the overall effect on the maneuverability of the flapping-wing aircraft.

3.       Line 19 How is the “better thrust-to-drag ratio” reflected in the study?

4.       There is a lack of statistical significance analysis in the performance comparison.

5.       Insufficient citation and comparative analysis of relevant research work.

6.       There is no figure depicting the experimental setup for stiffness testing.

Comments on the Quality of English Language

Minor editing of English language required

Reviewer 2 Report

Comments and Suggestions for Authors

Good paper, just few comments,

Review of The Aerodynamic Effect of Biomimetic Pigeon Feathered-Wing 2 on a 1-DoF Flapping Mechanism

In this paper entitled “The Aerodynamic Effect of Biomimetic Pigeon Feathered-Wing 2 on a 1-DoF Flapping Mechanism”, the authors performed experimental wind tunnel testing with a single-degree-of-freedom (1-DoF) mechanism for measuring models of wings of rock pigeons. Three wing models were created having different rigidity. Results were assessed under different flapping frequencies and undisturbed speeds.

Recommendation: This reviewer does not recommend publishing this paper in its current format. Minor revision includes some technical comments and manuscript arrangement needs to be done before publishing. However, the work done in this paper is pristine and it will be an adding value to the community.

Major Concerns:

1.     Section 2, flapping mechanism, I prefer to out all the mechanism constraints tabulated in a clear table to be easy for the readers, max. flapping angle, frequency ranges, max. torque…, etc.

2.     Figure 1, the mechanical system is not clear as it doesn’t show how the mechanism works. The main draw back of the mechanism is that it performs one degree of freedom flapping motion rather than dual plunging and pitching motion.

3.     In figure 9, the real wing obtained the same thrust force for static and free stream cases, please explain.

4.     As a matter of fact, the outer portion of the wing is responsible for the thrust generation and the inboard section id responsible for the lift generation. So, the outcomes of this research can be oriented to have a scientific fabrication of the tested wing to use the PLA for the inboard section and PETG to the outer section based on the author’s findings.

5.     I am also concerned about the accuracy of the results that have been presented. In all the cases investigated, irrespective of the tested case, the author didn’t show the actual (final) control gain values for each case in the form of table or so that can be useful for the readers.

6.     The uncertainty in measurements plays an important role is this type of testing, which in turn missed up in the manuscript. Please elaborate.

7.     Insufficient Analysis – the author presents experimental measurements the three wings. There are no other types of measurement to support / inform the obtained results. As a result, the discussion is very brief and I feel that the article raises lots of questions but gives few answers. In particular the dominant degree of freedom that affects the averaged lift and thrust generated is not sufficiently explained / demonstrated and there is no greater insight into the aerodynamic mechanisms causing these variations in the obtained results.

8.     It should be noted that, it will be more scientific if the authors presents the flapping frequency in terms of reduced frequency to judge the unsteadiness levels.

9.     The author seems to be unaware of a number of important references that have close connections to this work. The following references are a few that I am aware of. The authors should attempt to make connections between their results and the results presented in these papers.

a.    Mahardika, Nanang, Nguyen, Quoc Viet, Park, Hoon Cheol, “A pigeon-inspired design for a biomimetic flapping wing”, 7643, SPIE Smart Structures and Materials Nondestructive Evaluation and Health Monitoring, SS, 2010,  https://doi.org/10.1117/12.847354, DOI 10.1117/12.847354

b.    “Lift and Drag of Flapping Membrane Wings at High Angles of Attack”, M Y. Zakaria, David W. Allen, Craig A. Woolsey and Muhammad R. Hajj, AIAA 2016-3554, Flapping Flight Aerodynamics, Jun 2016, https://doi.org/10.2514/6.2016-3554

c.     Design Optimization of Flapping Ornithopters: The Pterosaur Replica in Forward Flight”, Mohamed Y. Zakaria, Haithem E. Taha and Muhammad R. Hajj, Published Online:8 Jun 2015, https://doi.org/10.2514/1.C033154

d.    Tanaka, S.; Asignacion, A.; Nakata, T.; Suzuki, S.; Liu, H. Review of Biomimetic Approaches for Drones. Drones 20226, 320. https://doi.org/10.3390/drones6110320

Reviewer 3 Report

Comments and Suggestions for Authors

It is a well-written manuscirpt with scientific merits. I have a few comments below to improve the quality of the paper. 

O Can you give more details how the cylindrical cam generate a sinosoidal flapping motion? Typically, people used four-bar-link mechanism; crank gear to drive a sinosoidal motion, but it seems like different or at least not clear besides some gear on the bottom in Fig. 1b. 

O Force data in figure 6 and 7 are quite noisy. Does your force sensor go through signal conditioner? Have you put any filter function? 

O For real bird wings, feathers have a special rachis structure to enhance its rigidity. I think that is why your real wing does not deflect much. Please check and cite this paper. https://doi.org/10.1098/rsif.2019.0267 

O Data availability statement is not applicable. But, it is more frank and can be checekd by third party if you open your data and design online. 

Round 2

Reviewer 1 Report

Comments and Suggestions for Authors

This paper has been revised, but it still has several deficiencies. After reviewing, I do not believe that this manuscript is suitable for publication in the journal. The content is insufficient and there are problems with the experimental design. Unfortunately, the author did not provide adequate explanation and supplementation in their response.

 1.        Our suggestion to set multiple groups of wind speeds is to increase the reliability of the results. There is a logical misunderstanding. If you are only comparing the mechanical and aerodynamic differences between feathers and flapping mechanisms and not evaluating the comparison between flapping mechanisms and real pigeons, the similarity between pigeons and flapping mechanisms does not need to be considered. Moreover, discussing results only at 16 m/s may not be representative because we do not know the pigeon's movement pattern at this speed, and perhaps it is not seeking extreme lift. After all, you have not evaluated other mechanical parameters, and the flapping mechanism does not have the maneuverability of a real pigeon.

 2.        If you are only interested in testing mechanical performance and aerodynamic behavior, do not limit yourself to only two wind speed variables. The reasons are the same as stated above.

 3.        Are the flapping mechanisms with different feathers the same? Also, there may be assembly errors with different wings, especially if there are many feathers involved. We are also unsure if the performance of the test prototype will change during the testing process. As you mentioned, artificial feathers exhibit more noticeable feather splitting, which is why we recommended significance analysis.

 4.        Regarding the gray background on the left side of Fig.3, I would need more information to understand its meaning.

Comments on the Quality of English Language

Minor editing of English language required

Reviewer 3 Report

Comments and Suggestions for Authors

Thanks for revising it. I think now it is clear how the cam shaft works in the design, which is interesting. Can you also provide CAD files as a supplementary material? That would be helpful for the readers. 

Author Response

It appears that providing the original CAD files is not a common practice, and unfortunately, we are unable to share the files publicly. However, in order to enhance your understanding of the mechanisms we have designed, we have included an exploded view of the main drive mechanism in the supplementary materials. We believe this visual aid will provide the readers with a clearer insight into the overall design and operational principles.

Round 3

Reviewer 1 Report

Comments and Suggestions for Authors

After two rounds of revisions, the author did not make substantial additions and improvements to the paper, causing many doubts still exist. And this work lacks sufficient novelty and originality to meet the standards of Biomimetics.

1) Experimental figures of bending test should be added in section 3.1

2) Supplementary experiment videos of Figure 5 need to be added to clearly indicate the wing deformation of feathers with different stiffness materials. Additionally, it is recommended to increase the flow field visualization experiment.

3) Figures 6-11 show incorrect units for lift or thrust.

4) The fabrication process of the 3D-printed artificial feather mentioned in Conclusion is not described in this study, which can be shown in drawings.

Comments on the Quality of English Language

Minor editing of English language required
